# Resistance of the Wheat Cultivar ‘Renan’ to Septoria Leaf Blotch Explained by a Combination of Strain Specific and Strain Non-Specific QTL Mapped on an Ultra-Dense Genetic Map

**DOI:** 10.3390/genes13010100

**Published:** 2021-12-31

**Authors:** Camilla Langlands-Perry, Murielle Cuenin, Christophe Bergez, Safa Ben Krima, Sandrine Gélisse, Pierre Sourdille, Romain Valade, Thierry C. Marcel

**Affiliations:** 1Université Paris Saclay, INRAE, UR BIOGER, 78850 Thiverval-Grignon, France; Camilla.Langlands-Perry@inrae.fr (C.L.-P.); murielle.cuenin@gmail.com (M.C.); christophe.bergez@free.fr (C.B.); safabenkrimapro@gmail.com (S.B.K.); sandrine.gelisse@inrae.fr (S.G.); 2ARVALIS Institut du Végétal, 91720 Boigneville, France; R.VALADE@arvalis.fr; 3Université Clermont–Auvergne, INRAE, UMR GDEC, 63000 Clermont-Ferrand, France; pierre.sourdille@inrae.fr

**Keywords:** bread wheat, Septoria tritici blotch, quantitative trait loci, *Stb20q*, strain specificity, resistance durability

## Abstract

Quantitative resistance is considered more durable than qualitative resistance as it does not involve major resistance genes that can be easily overcome by pathogen populations, but rather a combination of genes with a lower individual effect. This durability means that quantitative resistance could be an interesting tool for breeding crops that would not systematically require phytosanitary products. Quantitative resistance has yet to reveal all of its intricacies. Here, we delve into the case of the wheat/Septoria tritici blotch (STB) pathosystem. Using a population resulting from a cross between French cultivar Renan, generally resistant to STB, and Chinese Spring, a cultivar susceptible to the disease, we built an ultra-dense genetic map that carries 148,820 single nucleotide polymorphism (SNP) markers. Phenotyping the interaction was done with two different *Zymoseptoria tritici* strains with contrasted pathogenicities on Renan. A linkage analysis led to the detection of three quantitative trait loci (QTL) related to resistance in Renan. These QTL, on chromosomes 7B, 1D, and 5D, present with an interesting diversity as that on 7B was detected with both fungal strains, while those on 1D and 5D were strain-specific. The resistance on 7B was located in the region of *Stb8* and the resistance on 1D colocalized with *Stb19*. However, the resistance on 5D was new, so further designated *Stb20q*. Several wall-associated kinases (WAK), nucleotide-binding and leucine-rich repeats (NB-LRR) type, and kinase domain carrying genes were present in the QTL regions, and some of them were expressed during the infection. These results advocate for a role of *Stb* genes in quantitative resistance and for resistance in the wheat/STB pathosystem being as a whole quantitative and polygenic.

## 1. Introduction

Bread wheat (*Triticum aestivum*) is a staple food in many countries worldwide and is an economically important crop. Septoria tritici blotch (STB) is the most common disease of wheat and is caused by the ascomycete fungus *Zymoseptoria tritici* (formerly *Mycosphaerella graminicola*). It is responsible for high yield losses worldwide, ranging from 30 to 50% loss when environmental conditions are favourable to the disease’s development [1,2]. Disease control is generally undertaken with the application of fungicide treatments and/or cultivation of varieties carrying major resistance (*R*) genes. Neither of these methods, nor indeed their combination, are considered to be durable, thus necessitating constant renewal and research. Indeed, with ever-increasing restrictions on the use of chemical treatments in crops, a fully chemical approach does not seem to be the way to go. Another factor which encourages reducing the use of chemical treatments for this disease is their cost, which comes to more than 400 million euros a year in Europe [3]. *R* genes and their pyramiding appear as an excellent alternative. However, due to the existence of a gene-for-gene interaction between *R* genes and *Avr* avirulence genes in the fungus, the former are often overcome. Indeed, a single mutation in the *Avr* gene sequence can suppress recognition by the plant, stopping resistance mechanisms’ action [4]. Quantitative resistance is considered to be polygenic and to have a smaller effect than that due to *R* genes. It therefore imposes less selection pressure on fungal populations, making it more durable. Very few quantitative trait loci (QTL) or quantitative genes for resistance have been cloned, and the few that have are varied in terms of gene families and underlying mechanisms with cases of specificity and non-specificity [4,5,6,7,8,9,10,11,12,13,14]. The *T. aestivum*-*Z. tritici* pathosystem is considered to be primarily quantitative, though 22 major resistance genes in bread wheat have been described [15], the most recent being *Stb19* [16]. In controlled conditions as in the field, resistance to STB manifests itself as being quantitative, largely additive and with varying heritability. Brown et al. (2015) [15] counted 89 STB resistance QTL carrying regions, for a total of 167 individual QTL. Among these 89 regions are 62 QTL and 27 meta-QTL. Brown et al. (2015) [15] describe QTL as showing lower plant stage specificity than major resistance genes. These were not all detected at the same developmental stage, 27 at a seedling stage, 48 in adult plants and 14 for both of these stages. These regions are located on all wheat chromosomes but chromosome 5D, while chromosomes 3B, 6B, and 7B are more represented than others [15]. Moreover, several recent studies have reported additional QTL for resistance to STB [17,18,19,20,21,22], with r^2^ values ranging from 3.3% to 29.52%. Notably, Vagndorf et al. (2017) [17] detected QTL on chromosomes 1B, 2A, 5D, and 7A, with the QTL on chromosome 5D being the most effective of all four. These different studies show that STB resistance in wheat is complex, due to the combination of a large number of QTL with varying effects on phenotypes. No STB quantitative resistance genes have been cloned. However, major resistance gene *Stb6* was the first cloned gene specifying resistance to STB [23]. It encodes a wall-associated kinase-like protein, which detects the presence of a matching apoplastic effector. A second major resistance gene *Stb16q* was cloned in 2021, which confers broad-spectrum resistance against *Z. tritici* and encodes a cysteine-rich receptor-like kinase [24]. There is a gene-for-gene relationship between *Stb6* and *AvrStb6* [25], which is the avirulence gene which encodes a small secreted protein, the aforementioned apoplastic effector. This relationship entails specificity between cultivars carrying *Stb6* and strains carrying *AvrStb6*. Created in 1989 by INRA Rennes, Renan is a four-way hybrid resulting from a complex cross between cultivars Courtot, VPMxMoisson, Maris Huntsman, and Mironovskaia-808 (Doré and Varoquaux, 2006) [26]. Renan has the advantage of having very good baking quality and a high tolerance to the cold. Renan is resistant to a number of diseases such as cereal rusts, eyespot, septoria leaf blotch, and fusarium. These resistances result in part from the introgression of two chromosomal fragments from *Aegilops ventricosa* which carry the resistance gene to eyespot *Pch1*, on chromosome 7D, and resistance genes to rusts *Lr37*, *Yr17,* and *Sr38* [27,28,29] on chromosome 2A. Until the end of the 1990s, Renan had a good level of resistance to septoria leaf blotch. However, this resistance has been overcome by certain strains, even if Renan’s global disease resistance level remains reasonably good. Due to its disease resistance qualities, relative shortness and reasonably good competitive value against weeds, Renan is a popular choice for organic farming [30]. Chinese Spring is the reference genome for wheat and has been used in a great number of studies [31]. It was the first wheat to have its genome fully sequenced and assembled [32,33], and that sequence remains the most contiguous wheat assembly to date [34]. Moreover, it is sensitive to a wide array of biotic and abiotic stresses, including STB. This means that it is ideal as a sensitive parent for a bi-parental population with a resistant parent such as Renan [35]. One thing to bear in mind when working with Chinese Spring is that it carries resistance gene *Stb6* and is therefore resistant to isolates carrying the corresponding avirulence gene *AvrStb6* [25]. The aim of this study is to better understand quantitative resistance to STB and the isolate-specificities of QTL. We hypothesize that these specificities could be due to minor gene-for-minor gene interactions, such as those that have already been suggested in other works, in the cases of barley—barley leaf rust [36,37], barley—barley leaf stripe [38], potato—*Phytophthora infestans* [39] or indeed pepper and potyviruses [40]. Further knowledge of these interactions would give us better understanding of the durability of quantitative resistance.

## 2. Materials and Methods

### 2.1. Plant and Fungal Materials

The plant material used in this study consisted in a population of 236 Recombinant Inbred Lines (RILs; F6 generation) obtained from a cross between wheat cultivars Renan and Chinese Spring. Hereafter, this population will be referred to as RxCS. The fungal strains used in this study are INRA09-FS0813 and INRA09-FS0732, which we will refer to hereafter as I05 and I07, respectively. These two strains were isolated from leaves of the cultivar Soissons collected from the same field in 2009 at Thiverval-Grignon in France. They are both virulent on *Stb6*. They were chosen for this study out of a panel of eight different strains as they were both pathogenic on Chinese Spring and showed contrasted pathogenicity on Renan. I05 can infect Renan, while I07 cannot.

### 2.2. Experimental Setup and Procedure for Pathology Assays

#### 2.2.1. Data Sets

Due to the size of the population, it could not be phenotyped in its entirety over one experiment. It was therefore divided into two equal sets, which were tested in 2017 with two replications. All phenotypic traits were evaluated for this set. To corroborate results, 148 individuals were chosen randomly out of the original 236, and tested over three replications in 2018. For this data set, PYC and NBS (explained later on) were not evaluated. For every experiment, Renan and Chinese Spring were used as controls.

#### 2.2.2. Culture Conditions

For each individual in the population, five seeds were sown per pot filled with Floradur B (Floradur Pot Medium) potting soil (NPK 14, 16, 18 kg∙m^−3^) (Floragard Vertriebs-GmbH, Oldenburg, Germany). The pots were split into trays with 15 pots per tray, for a total of 8 trays per experiment. Before inoculation, plants were cultivated in a climate chamber with a 16h photoperiod, hygrometry fixed at 70% and temperatures at 16 °C night and 20 °C day. Post-inoculation, plants were placed in a climate chamber with a 16h photoperiod, hygrometry fixed at 90% by day, 80% by night and temperatures of 22 °C during daytime and 18 °C at night-time. In the climate chambers, light conditions were maintained at 300 µmol∙m^−2^∙s^−1^ with eight neon tubes (Osram Lumilux L58W/830 placed 40 cm above the trays (OSRAM GmbH, Munich, Germany).

#### 2.2.3. Inoculum Preparation

Inocula were prepared from strains conserved at −80 °C as spores in a 70% water and 30% glycerol mix. The strains are precultured 10 days prior to inoculation in 10 mL of a YPD liquid culture medium (1% yeast, 2% bacto-peptone, 2% glucose). The precultures were kept in a growth chamber for 6 days at 17 °C and a hygrometry of 70% under agitation (140 rpm). Each preculture was then grown in a petri dish (Ø 90) on a PDA (potato dextrose agar) solid culture medium with 50 to 80 µL of inoculum. The day of the inoculation, 150 mL of inoculum were prepared from these cultures, each with a concentration of 1.106 ± 0.1.106 spores∙mL^−1^. Lastly, one drop of Tween 20 was added per 15 mL of inoculum to insure adherence of the inoculum to the leaf surface.

#### 2.2.4. Inoculation

Plants were inoculated 16 days after sowing. The day before the inoculation, only three plants out of a maximum of five were kept per pot. On the first true leaf (generally 3 to 5 cm from the base) of each plant, a surface of 7.5 cm in length was marked out with two black felt tip lines. The inoculum was spread out on this surface using a square-tipped flat paintbrush in six passages (3 times back and forth). Once all the pots of a tray were inoculated, the procedure was carried out a second time. The paintbrush was dipped into the inoculum before each set of six passages. After inoculation, each tray was watered and covered with transparent polyethylene bags. The bags create a water-saturated atmosphere, which encourages infection [41,42]. The bags were removed after three days, a 72 h incubation period being the time it takes for the fungus to reach the mesophyll, which is necessary to the rest of the colonisation process [43]. To optimise conditions for the survival of the inoculated leaf and to homogenise the quantity of light received by each leaf, new leaves were cut 2 to 3 cm above the first node 10 days post-inoculation (dpi).

### 2.3. Evaluation of Phenotypic Traits

#### 2.3.1. Visual Evaluation of Symptoms

The leaf area marked out with black felt tip was visually evaluated at 14, 20, and 26 dpi. The percentage of the surface which was green, necrotic, and sporulating was evaluated. The sporulating area is defined here as the area which presents pycnidia, regardless of density, colour, or size. The values for sporulating area at 14, 20, and 26 dpi were used as phenotypic traits in linkage analyses and are referred to as S14, S20 and S26. The chlorotic area was not evaluated because it is deductible from the green and necrotic areas. These notations were used to calculate AUDPCs (Area Under the Disease Progress Curve) for the green, necrotic, and sporulating areas (AUDPCG, AUDPCN, and AUDPCS, respectively) [44]. The formula for calculating an AUDPC is as follows:$$\mathrm{AUDPC}=\frac{\sum^{\;}\left[\left(t_{i+1}-t_{i}\right)\times \left(y_{i}+y_{i+1}\right)\right]}{2}$$

With:

*t_i_*_+1_−*t_i_*: number of days between two notations (6 days)

*y_i_*: percentage of green, necrotic, or sporulating area at day *i* (for AUDPCG, AUDPCN and AUDPCS respectively)

*y_i_*_+1_: percentage of green, necrotic, or sporulating area at day *i*+1 (for AUDPCG, AUDPCN and AUDPCS respectively). 

#### 2.3.2. Pycnidia Counting by Image Analysis

The procedure followed and macros used were developed by Stewart and McDonald (2014) [45] and improved by Stewart et al. (2016) [46]. Images were obtained by scanning the inoculated part of the leaf. Firstly, for sample identification, A4 pages are generated using a Linux supported macro. The page is divided into eight sections, each carrying a QR code which is specific to a sample. For each sample the three leaves are glued inside the corresponding section. Pages are then scanned using a CanoScan 9000F MarkII scanner (resolution = 1200 dpi, luminosity = contrast = 0) and the resulting images are saved as .tiff files. Pycnidia density was evaluated using a macro in ImageJ [47]. This macro is able to quantify several traits which are the percentage of the leaf surface covered by lesions or by necrosis only (PLACL or PLACN respectively), the total number of pycnidia per sample, the size of the pycnidia and the grey value of the pycnidia. For this study, only the total number of pycnidia per sample was taken into account (one sample being the three inoculated leaves for each strain). Parameters within ImageJ were adjusted for each image as it was not possible to use the exact same parameters for each sample, depending on the colours of the leaves. 

Pycnidia density was calculated for each sample using the following formula:$$\mathrm{PYC}=\frac{n_{pycnidia\;total}}{N_{total}}$$

With:

*n_pycnidia total_*: total number of pycnidia in the sample.

*N_total_*: total necrotic surface of the sample.

#### 2.3.3. Quantification of Sporulation

Sporulation was quantified with the use of the particle size & shape analyser Occhio Flowcell FC200S+HR (Occhio s.a., Angleur, Belgium). This tool is controlled by a computer which, with the help of image analysis, can precisely count particles while evaluating their size and shape. It is used alongside the Callisto software which controls the quality of the analysis. The user interface allows one to control various parameters such as light, image resolution and particle selection settings (Table 1). The latter is particularly important as it means that any particle that is not a pycnidiospore can be excluded from the analysis. Preparation of the samples for analysis is done the day after the last visual notation (27 dpi). The inoculated area of the leaf is cut out. The three leaves from each genotype are placed into a 15 mL Falcon tube containing 0.75 mL of osmosed water and two pieces of blotting paper (10.5 cm × 0.5 cm). The blotting paper maintains a water-saturated atmosphere within the tube which favours the extrusion of cyrrhi from pycnidia, and thus sporulation [48]. Tubes are then placed in a growth chamber for at least 18 h. On the next day, 5 mL of an 80% water 20% glycerol mix is added to each tube. The tubes are then gently agitated so that the spores present on the surface of the leaves are transferred to the liquid phase. The leaves are then removed from the tubes and mounted on a white paper sheet for scanning. The tubes containing pycnidiospores suspended in 5.75 mL water/glycerol are kept at −20 °C awaiting analysis. Before the analysis, each tube is homogenized after the addition of a drop of Tween 20 to the mix. 0.9 mL of the mix are used for each analysis. As each sample is passed through the particle counter, an image control is used to verify that all particles are counted independently. Over 350,000 particles mL^−1^, the initial mix is too concentrated in particles thus it becomes necessary to dilute the sample and repeat the procedure. The particle counter is rinsed with osmosed water between each sample. 

For each sample, the total number of pycnidiospores was extracted from data obtained from this particle counter. The total number of pycnidiospores was used to calculate the number of pycnidiospores per pycnidium after image analysis.

The number of pycnidiospores per pycnidium (NBS) was calculated using the formula:$$\mathrm{NBS}=\frac{n_{pycnidiospores\;total}}{n_{pycnidia\;total}}$$

With:

*n_pycnidiospores total_*: total number of pycnidiospores extracted from the sample

*n_pycnidia total_*: total number of pycnidia in the sample.

### 2.4. Statistical Analysis of Phenotypic Data

The obtained data sets were analysed with the R software [49], for each trait an analysis of variance (ANOVA) was performed with the following model:$$Y_{ij}\;=\;µ\;+\;I_{i}\;+\;r_{j}\;+\;Ir_{ij}\;+\;\varepsilon i_{j}$$

With *Y_ij_* the trait which is being studied, µ the mean value for this trait, *I_i_* the individual genotype, *r_j_* the replication, *Ir_ij_* the interaction and *ε_ij_* the residual. For the following analyses, *Ir*_ij_ was included in the residual.

The following hypotheses were verified after the variance analyses.

$\varepsilon \;\sim \;N\left(o,\sigma^{2}\right)$ →$cov\left(\varepsilon ,\varepsilon^{\prime }\right)=0\;$ Homoscedasticity (homogeneity of var(ε))

Broad sense heritability is defined by the following formula:$$H^{2}\;=\;\frac{\sigma_{g^{2}}}{\sigma_{g^{2}}\;+\;\sigma_{e^{2}}}$$

With $H^{2}$ the heritability, $\sigma_{g^{2}}$ genotypic variance and $\sigma_{e^{2}}$ residual variance. 

The correlation between traits was also studied using the Bravais–Pearson correlation.

### 2.5. Genotyping RxCS

The wheat population was genotyped on two different single nucleotide polymorphism(SNP) arrays, the Breedwheat Affymetrix Axiom 410K array [50,51] and the Illumina Infinium iSelect Wheat 90K array [52].

#### 2.5.1. Axiom 410K

In total, 429 individuals from the RxCS population were genotyped on the Axiom 410K array in two sets of 282 and 147. Among these, 236 individuals were randomly selected for further phenotyping. DNA extraction and genotyping were performed by the Gentyane platform (INRAE, Clermont-Ferrand, France).

#### 2.5.2. ISelect 90K

For genotyping on the iSelect 90K array, 159 individuals chosen randomly among RxCS and both parental varieties were genotyped. DNA extraction and genotyping were performed at TraitGenetics GmbH (Gatersleben, Germany).

### 2.6. Genetic Analyses

#### 2.6.1. Construction of an Ultra-Dense Genetic Map

Markers were re-named so as to be tagged with their SNP array of origin and the chromosome they most likely mapped to as this made following steps more straightforward. There are 13,462 markers in common between the 90K array and the 410K array. These were considered to be distinct for map construction. Out of a total of 409,695 SNP, were kept for the map building file only those that were polymorphic and for which the information from both parents was available, leading to a total of 194,630 SNP. Additionally, only 142 individuals’ information was used for map construction as for 17 individuals, genotyping on the 90K had done poorly. 

The file comprising all polymorphic markers from the TaBW410K and the iSelect90K arrays was input into the Multipoint ultra-dense software developed by MultiQTL Ltd. (Haifa, Israel) at Haifa University in Israel. This software allows ultra-dense genetic maps to be built and is based on the “twin algorithm” [53]. A stringent filter for missing data was applied, all SNP with over 5% missing data were filtered out. Then, a filter to correct any potential segregation distortion was applied; it was less stringent, allowing a Chi2 up to 9.5. After the first clustering, linkage groups, which belonged to the same chromosome, were merged and the results from Multipoint were transformed with the Kosambi mapping function [54]. The linkage map was graphically visualized with Mapchart V2.3 [55].

#### 2.6.2. Linkage Analysis

A linkage analysis was carried out using the R/qtl software [56] version 1.42-8. This analysis included for each trait an initial Simple Interval Mapping (SIM), followed by a Composite Interval Mapping (CIM). Analyses were performed replication by replication and set by set (sets only concerning the first lot of phenotypic data). For SIM, 1000 genome-wide permutations were used to calculate the significant logarithm of odds (LOD) threshold. Only QTL that showed *p*-values < 0.05 were considered significant. The CIM was carried out with the SNP with the highest LOD at QTL peaks used as a covariate. When there were several significant QTL detected, the CIM was recalculated with two covariates, however, this never led to any extra detections. QTL intervals were evaluated with the LOD support interval with a drop in LOD of 1 and the “expandtomarkers” argument as true. QTL effects were calculated with the “effectplot” and “effectscan” functions. Possible epistatic interactions between QTL were looked into using the “addint” function. 

#### 2.6.3. QTL Gene Content

The gene content of the QTL regions was analysed using the IWGSC RefSeq v1.1 annotation, which is anchored on the IWGSC RefSeq v1.0 assembly (both available at https://wheat-urgi.versailles.inra.fr/; accessed on 3 December 2021). We specifically searched for the content in wall-associated kinases (WAK), and nucleotide-binding and leucine-rich repeats (NB-LRR) type genes using the IWGSC’s 2018 work on manually curated gene families [32] and we searched through the annotated genes using the keyword “kinase”. By using the Wheat Expression Browser powered by expVIP (http://www.wheat-expression.com/; accessed on 3 December 2021) [57,58] we were able to access transcriptional data for each of the listed genes in a kinetic of STB infected plants. The data available was for the cultivar Riband at 1, 4, 9, 14, and 21 dpi infected with fungal strain IPO323 [59]. This cultivar is regarded as a good positive control for *Z. tritici* infection as it is highly susceptible to the disease, moreover, it does not carry the major resistance gene *Stb6*, unlike Chinese spring for instance [60]. To compare with known STB resistance genes, we included the expression data for *Stb6* and *Stb16q*, respectively known as *TraesCS3A02G049500* and *TraesCS3D02G500800* in the RefSeq v1.1 annotation.

## 3. Results

### 3.1. Description of Phenotypes

Two phenotypic data sets were acquired on the Renan × Chinese-Spring population for the isolates I05 and I07. The first data set was collected in 2017 on 236 RILs with two replications, and the second data set was collected in 2018 on 148 RILs with three replications. Overall, distributions of phenotypic traits within the population are similar between replications, and Renan is consistently more resistant than Chinese Spring (Appendix A). The second replication from the 2018 data set of isolate I05 stands out as being quite different from the others, in particular for S26 and AUDPCS. For this replication, the level of infection was lower than for the other replications and heterogeneous, suggesting that infection failed. Therefore, the following analyses were performed excluding the data from this particular replication, leaving only two replications for the 2018 data set of isolate I05. It is also important to notice that distributions are not bi-modal; phenotypes follow a continuous distribution indicating the presence of several genes with quantitative effect. Only for S26 in 2018, for both isolates I05 and I07, does the distribution not look strictly continuous, but rather skewed towards susceptibility. Finally, the parental phenotypes do not mark the limits of the distribution. Transgressive individuals can be observed for the great majority of studied traits indicating that despite Chinese Spring being susceptible to both fungal strains, it can carry small effect resistance QTL. Correlation coefficients were calculated from mean values between replications using the Bravais–Pearson correlation coefficient (Figure 1). For these analyses, the 2017 sets were analysed together. Overall, AUDPCN, AUDPCS, S20, S26 and PYC (only 2017 data) were all strongly correlated. AUDPCG was poorly correlated to the other traits, except for the 2018 I07 data, where it is strongly negatively correlated to the other traits. Overall, the S14 and NBS traits were poorly correlated to the other traits. This is certainly due to the low values and low variability in S14 data, and the low reproducibility in NBS data. These results suggest a potential common genetic base for all of the studied traits.

The ANOVA results show that the individual genotype systematically had a significant effect on the phenotype in all cases but I05_2017/1 S14 and NBS, and I07_2017/2 S14 (Table 2). The replication overall had a strong effect on phenotypes, except for the I07_2017/2. Subsequent linkage analyses were therefore carried out replication by replication. Broad-sense heritability was calculated for all traits and was systematically higher for I07 data than for I05 data (Table 2). As the ANOVA assumptions are not respected in all cases, low heritability value does not necessarily indicate that the trait will not lead to the detection of resistance QTL.

### 3.2. An Ultra-Dense Genetic Linkage Map

We genotyped 159 RILs from the RxCS population with both TaBW410K and iSelect90K arrays. These two arrays share 13,670 common markers, but comparison of parental genotypes between arrays for these common markers revealed an average divergence of 11.1% for both cultivars. Due to these divergences common markers between both arrays were given different names and considered separately for map construction. Overall, only markers which were polymorphic between both parents and presented no missing data, lack of amplification or heterozygosis for either parent were kept, leading to a total of 183,773 usable markers from the TaBW410K array, representing 43.4% of the original data set, and a final list of 10,857 markers for the iSelect 90K array, representing 13.3% of the original data set. Consequently, for this population, the rate of polymorphic markers was more than three times higher on the TaBW410K than on the iSelect90K. The 194,630 SNP markers available for map construction were all re-named to show which array they came from and onto which chromosome they were expected to map based on best BLAST values on the reference genome (Chinese Spring). This enabled us to associate linkage groups and corresponding chromosomes more easily when building the map. From the 159 individuals genotyped with both TaBW410K and iSelect90K arrays, 17 individuals were discarded because of their high number of missing data from the iSelect90K genotypes. Moreover, applied filters deleted 13.4% of the markers, leaving us with a matrix of 142 individuals x 168,522 high quality markers for map construction. After the initial clustering, 25 linkage groups were obtained. Those that corresponded to different parts of the same chromosome were merged. The map we obtained comprised 21 linkage groups, each corresponding to a chromosome (Table 3). It is composed of 5357 genetic bins or unique positions, for a total of 148,820 markers and covers a total genetic distance of 4277 centiMorgans (cM). Appendix A provides the complete map. Of all mapped markers, 3.54% were skeleton markers, each representing a genetic bin. Of the original number of available markers from the SNP arrays, 30% were mapped, and of the markers chosen for mapping after all filters, 78% were mapped. Sub-genomes A and B carry more SNP than sub-genome D. When working with the full set of markers, sub-genome D represents only 16.50% of this latter set, while A and B sub-genomes represent 42.78% and 40.72%, respectively. However, sub-genome B has a shorter genetic length than either of the other two. The great majority of mapped markers are from the Breedwheat Axiom 410K array; indeed, these SNP represent 94.73% of sub-genome A, 94.61% of sub-genome B and 97.91% of sub-genome D. The 90K array is particularly underrepresented on the D sub-genome with markers making up a minimum of 0.84% on chromosome 4 and a maximum of 3.73% on chromosome 6. The distance between consecutive markers ranges from 0.35 cM to a maximum of 21.54 cM. However, a gap this size is exceptional. Indeed overall, consecutive markers are quite close together, with the average distance between consecutive markers being 0.74 cM in sub-genome A, 0.69 cM in sub-genome B and 1.07 cM in sub-genome D. The third quartile in the distribution of the distance between consecutive markers is 0.72 and 0.71 cM for sub-genomes A and B respectively, while it is slightly higher for sub-genome D with a value of 1.1 cM. In sub-genome A, six gaps in the map are larger than 10 cM on chromosomes 3A, 4A, 5A and 7A. In sub-genome B, only one gap is larger than 10 cM and is on the long arm of chromosome 3B. Moreover, sub-genome D has four gaps larger than 10 cM on chromosomes 1D, 6D (twice), and 7D.

The comparison between the order of the markers on the genetic map and their assumed physical position shows that, overall, the genetic map follows the assumed physical positions very well (Figure 2). The exception is chromosome 4D, which presents a cluster of markers at the end of the linkage group, which would have been expected to be found in the short arm according to their assumed physical positions (Figure 2). The ratio between genetic and assumed physical positions is also interesting to comment on as it clearly shows the structure of the different chromosomes with SNP distribution following a neat sigmoidal pattern. Indeed, the position of centromeres is marked out by the recombination suppression surrounding them on all chromosomes (Figure 2). 

### 3.3. Mapping QTL for Resistance

The analyses of the 2017 and 2018 datasets lead to the detection of several QTL on eight different chromosomes. Among these, only three were detected more than once throughout the analyses, we will therefore focus on these three. The complete linkage analysis results can be found in Appendix A. The three robust QTL were detected on chromosomes 1D, 5D and 7B (Table 4 and Table 5). The parent carrying the resistant allele is Renan for all three QTL. The QTL however, do not impart resistance to the same strains. Indeed, *Qstb.renan-1D* imparts resistance to I05 while *Qstb.renan-5D* imparts resistance to I07 and *Qstb.renan-7B* is effective against both strains. *Qstb.renan-7B* explained up to 32% of phenotypic variation with a mean r^2^ value of 20% when detected with the 2017 datasets and up to 38% with a mean value of 21% when detected with the 2018 datasets. *Qstb.renan-1D* was detected the fewest times out of the three repeatable QTL and explained the least phenotypic variation, between 6 and 13.5% on average, to a maximum of 15%. This QTL was not detected with the 2017/2 dataset (Figure 3B). Finally, *Qstb.renan-5D* explained up to 35.5% of phenotypic variation with a mean r^2^ value between 15.5 and 26%. We did not identify any QTL trait specificity in the various linkage analyses, as was to be expected regarding the correlations between traits. The traits that led to the most detections are S26, AUDPCS and PYC. The traits associated with the highest r^2^ values overall are AUDPCN, AUDPCS, S26, and PYC. Concatenated results between the 2017 and 2018 data sets result firstly in *Qstb.renan-7B* that is 50.86 Mb long, spanning 64.62 cM. Secondly, *Qstb.renan-1D* is 4.28 Mb long, covering 17.79 cM and carrying 5 SNP with known physical positions. Finally, *Qstb.renan-5D* is 312 Mb long, spanning 28.58 cM. The physical intervals were estimated using markers in the QTL with known positions mapping to the chromosome in question. These numbers take into account every detection of each QTL, and therefore include the least precise detections; this explains, in part, the very large intervals. We however do detect maximum LOD score peaks in the same area throughout the analyses (Figure 3 and Figure 4). No significant interactions were detected between QTL.

### 3.4. Gene Content of the QTL

There are two cloned major resistance genes to STB in wheat to date. The first is *Stb6*, which encodes a wall-associated kinase-like protein and detects the presence of a matching apoplastic effector [23]. The second is *Stb16q*, which encodes a cysteine-rich receptor-like kinase and confers broad-spectrum resistance against *Z. tritici* [24]. With this in mind, along with the very large QTL intervals we are working with, we opted to focus on three particular gene families when analysing gene content within the QTL intervals. These families are wall associated kinases (WAK), nucleotide binding leucine-rich receptors (NB-LRR) type genes and genes carrying a kinase domain, all of which have often been associated with disease resistance [24,61].

*Qstb-renan-1D* holds a total of 141 annotated genes. Of the 141, 25 are NB-LRR-type genes, while 5 are WAKs and one has a kinase domain. *Qstb-renan-5D* holds 1981 annotated genes, none of which are NB-LRR-type genes. It holds 12 WAKs and 22 genes with kinase domains, two of which are projected to be receptors or receptor-like, these are *TraesCS5D02G166400* and *TraesCS5D02G181500*. *Qstb-renan-7B* holds 616 annotated genes. This QTL holds no NB-LRR type genes, one WAK, and one gene with a kinase domain. Table 6 presents the details concerning the NB-LRR, WAK and kinase domain carrying genes in the QTL intervals.

With the data extracted from expVIP (http://www.wheat-expression.com/; accessed on 3 December 2021), we were able to see that for a majority of the genes, there was no expression in the inoculated plants. However, for 35 of them, we were able to identify expression patterns over the course of the infection (Figure 5). What we can observe with this data is that for the two known resistance genes *Stb6* and *Stb16q*, expression is maximal at 9dpi, this is also the case for some of the WAK and NB-LRR type genes in the QTL intervals. At 9dpi, the infection is still in the biotrophic phase but at the onset of switching to the necrotrophic phase [62]. For *Qstb-renan-1D*, these are *BST_chr1D_nlr_113*, *TaWAK40_1D-gene*, *TaWAK41_1D-gene*, *TaWAK42_1D-gene*, and *BST_chr1D_nlr_102*. For *Qstb-renan-5D*, these are *TaWAK355_5D-gene*, *TaWAK356_5D-gene*, *TaWAK358_5D-gene*, and *TraesCS5D02G081700*. For the two receptor or receptor-like kinases in *Qstb-renan-5D*, there was no expression. The only gene with any expression in our list for *Qstb-renan-7B* was *TraesCS7B02G466300*, which had low expression overall, but it was maximal at 1 dpi.

## 4. Discussion

### 4.1. An Ultra-Dense Genetic Map Built from Two SNP Arrays

In this study, we built an ultra-dense genetic linkage map for bread wheat from the RxCS population. The map we obtained boasts 5357 unique positions for a total of 148,820 mapped markers. At present, this is the most densely marked genetic map built from a single segregating population. It does contain eleven gaps that are larger than 10 cM, which could hinder the detection of QTL in these regions. Moreover, for the most part the assumed physical position of the markers and their order in the genetic map corresponds well. There is a cluster of markers on chromosome 4D that mapped on the long arm of the chromosome while their physical position is assigned to the short arm. This could be due to chromosomal rearrangements between Chinese Spring and Renan, although the D sub-genome is not the most prone of the three to chromosomal rearrangements [63,64]. The Renan–Chinese Spring genetic map built by Rimbert et al. (2018) [65] corroborates the marker ordering we find with this map for chromosome 4D and indeed for the rest of the genome (Appendix A). Finally, a recent refined assembly of Chinese Spring [33] includes several major corrections at the 4D long arm telomere, suggesting that the misplacement is indeed due to errors in the original assembly.

Two studies, which used only Illumina Infinium iSelect 90K markers, provide a good base for comparison with the map we built here. The first is that of Wang et al. (2014) [52] which was the first built with this array. They built a consensus map using six doubled haploid mapping populations; the map contains 40,267 markers distributed in 5564 genetic bins. In the genetic maps that they built for their various mapping populations, the majority of markers were to be found in the A and B sub-genomes while the D sub-genome represented a mere 15% of mapped markers, consistent with what we found in this study. The order of the markers in our map is for the most part consistent with that in Wang et al. (2014) [52] (Appendix A). The second study we used for comparison is that of Wen et al. (2017) [66]. They built a consensus map comprising 29,692 markers distributed in 8960 bins; it was built with the maps of four mapping populations genotyped with the Illumina Infinium iSelect 90K array and five maps from previous reports. Firstly, all four of their maps consistently have a lower number of markers on the D sub-genome than on sub-genomes A and B, with the percentage of total markers on the D-sub-genome ranging from 7.64% to 12.8%, this corroborates with our results and other studies which show lower diversity in the D sub-genome [66,67,68]. The total number of markers on each map is 10,986 for 2840 bins, 11,819 for 3242 bins, 9824 for 3198 bins and 14,862 for 3460 bins. The map presented here carries a total of 148,820 markers for 5357 bins. It is therefore very densely covered and rivals the Wang et al. (2014) [52] consensus map in terms of unique genetic positions despite being a map built from a single mapping population. Its particularity lies in the fact that two different SNP arrays were used to construct it, the TaBW410K Breedwheat array and the Illumina Infinium iSelect Wheat 90K array. The final RxCS map contains 95% of markers from the TaBW410K Breedwheat array and only 5% from the second array, this can in part be explained by the fact that the markers constituting both arrays were not chosen following the same strategies. For the TaBW410K Breedwheat array, a particular effort was made to target polymorphic markers, notably in modern wheat cultivars. Though the Infinium iSelect Wheat 90K array SNP in the RxCS genetic map are fewer compared to those of the Breedwheat array, they are present throughout the map on sub-genomes A and B (Appendix A). Both arrays therefore contribute to the map’s construction and structure (Appendix A). The 13,670 SNP markers, which are supposedly common to both arrays, do not often appear in the same genetic bins, though they do for the most part appear at close genetic positions. Indeed, out of 13,670, only 33 markers appear in the same bin as their counterpart in the other array. The average distance overall between two paired markers is 3.19 cM, while the median value is 2.27 cM. In one exceptional case, that of BS00044983_51, the markers of the pair do not appear on the same chromosome but on the homoeologous chromosomes 2B and 2D. The largest gap between a pair of markers, which are mapped on the same chromosome, is 21.1 cM and is the case for marker BobWhite_c12355_1548. In total, 27 marker pairs are more than 10 cM apart. This could be due to error rates between genotyping experiments, slightly different scoring methods for SNP calling between the arrays or indeed duplications in the sequence. All these results in addition to the very low number of mapped markers from the iSelect Wheat 90K array shows that the TaBW410K Breedwheat array performs better as a base for mapping.

As of yet, this is the only map which carries markers from both of the SNP arrays previously mentioned. As such, this map can be used as a “bridge” to compare maps from studies using only one or the other SNP array.

### 4.2. Phenotypic Traits Involved in the Resistance of Renan to STB

All of the experiments presented here were carried out in controlled conditions at the seedling stage. This simplifies phenotyping as, in field conditions, other diseases with similar symptoms to STB can infect crops. Septoria nodorum blotch caused by fungal pathogen *Parastagonospora nodorum* in particular can lead to confusion [69]. Other external factors such as environmental variations can also greatly impact field conditions, compromising experiments’ success [70]. Another reason for choosing to work in controlled conditions is that quantitative interactions can be highly variable within and between experiments [44,45,71], controlled conditions allow one to reduce background variation that might exist in field conditions, although it is not completely suppressed. As the values for S26, AUDPC for green area, necrotic area, and sporulating area were all generated using data from visual symptom assessments, it was necessary to reduce background noise as much as possible, this included always having the same person perform the assessments within a triplicated or duplicated experiment. These precautions allowed us to obtain data, which though not necessarily normally distributed or statistically repeatable according to ANOVA results, enabled us to identify reliable QTL when comparing overall results. Precision phenotyping with image analyses has already been used successfully for studying the *Z. tritici*/wheat interaction [46]; it is suggested that pycnidia counting and size evaluation could be helpful in evaluating the epidemic potential of fungal strains and that the host can influence these traits. Yates et al. (2019) [21] identified previously undetected loci for quantitative resistance to STB using the ImageJ macro for four phenotypic traits evaluated on a GWAS panel of 335 European winter wheat cultivars infected with a natural highly diverse *Z. tritici* population. In our case, though the PYC trait corresponding to pycnidia density did enable us to detect QTL, these were not different to the ones we were already able to identify using the visual assessment data. Precision phenotyping with a particle counter has also previously been used successfully in the *Z. tritici*/wheat interaction. Boixel, et al. (2019) [72] evaluated the number of spores in a sample and they also evaluated spore size, shape and melanisation. They showed that for spore counting, the particle counter is a good means of gaining in accuracy and with the study of morphological variation between spores, they provide insight into more phenotypic traits that could be accessed by using a particle counter. Particle counters have been used in the study of the *Melamspora larici-populina*/poplar interaction [73], the *Phytophthora infestans*/potato interaction [74] and the two main species of the ascochyta blight complex of pea/pea interaction [75]. In this latter case, the particle counter’s image analysis capacities were used to evaluate spore length. This type of phenotyping offers a large diversity of potential phenotypic traits, which could lead to the identification of novel QTL, however, much as for PYC, the sporulation trait NBS did not yield the results we had hoped for, however it did confirm the QTL that we could already identify with our visual assessment data. These traits have previously led to the detection of QTL that were different to those detected with visually acquired data, however this was in the context of a GWAS with a diversified panel of cultivars [21].

### 4.3. Quantitative Resistance Durability Is a Multi-Layered Issue

Our objective was to identify regions in the *T. aestivum* genome, which could carry genes that contribute to quantitative resistance to STB. The continuous distribution of phenotypes that we observed in our data is indeed indicative of a quantitative and polygenic type of resistance (Appendix A). We were able to identify three robust QTL on chromosomes 1D, 5D, and 7B, each explaining between 6% to 26% of the phenotypic variation on average. These quantitative and polygenic attributes contribute to resistance durability for several reasons. Firstly, quantitative resistance, in theory, exerts less selection pressure on pathogen populations, meaning that the latter are less likely to adapt in the short term [76]. Secondly, the polygenic nature of quantitative resistance provides more arguments for durability and is illustrated in the present case by the identification of three different QTL. Polygenic resistance is more durable as it is a combination of factors which provides resistance, rendering adaptation to each of these factors more complex [77]. This can be illustrated with studied cases of viral plant pathogens where the higher the number of mutations required for virulence is, the lower the probability of adaptation gets, and therefore polygenic resistance is more durable than monogenic [78,79,80]. In the case of the *Capsicum anuum*/*Potato virus Y* in particular, it was shown that polygenic resistance breakdown is slower than monogenic, indeed the virus required a step-by-step selection for virulence, first towards major resistance genes, then towards the QTL and major gene combinations [80,81]. Brought back to our study case, as a combination of resistance QTL is efficient against a fungal strain, the adaptation to one of the QTL will not render overall resistance void. This is illustrated by the strain specificities of the identified QTL; i.e., though a strain has adapted to a specific QTL, this same QTL remains effective against the other strain. *Qstb-renan-1D* was detected only with fungal strain I05, while *Qstb-renan-5D* was only detected with fungal strain I07. Finally, *Qstb-renan-7B* did not discriminate between the strains. We therefore have different QTL combinations that are efficient against different fungal strains. This entails no specific strain selection as the selection pressure is in a way diluted between strains [76]. It has been previously suggested that either pyramiding broad-spectrum factors or using host genotypes carrying narrow-spectrum resistance QTL could minimize resistance erosion [82] and that combining specific QTL that can complement each other can slow selection of one particular pathogen strain [82] increasing the durability of quantitative resistance. 

Though there are many arguments in favour of durable quantitative resistance, it is not necessarily the panacea. There are pathosystems where quantitative resistance has been overcome through pathogen adaptation. In the case of the aforementioned pepper/*Potato virus Y*, quantitative resistance alone was not sufficient to halt adaptation, emphasis was put on the possibility of implementing cultivar mixtures [83]. In the apple/*Venturia inaequalis* pathosystem use of broad-spectrum quantitative resistance was shown to present a risk of encouraging the emergence of generalist pathogen populations [84]. In the wheat/*Puccinia recondita* f. sp. *tritici* pathosystem, it has been shown that partial resistance can be eroded when natural selection is high [85]. Another example is the grapevine/*Plasmopara viticola* interaction, where partial host resistance rapidly selected for pathogens with higher virulence [86]. These different examples show that though quantitative resistance is considered to be durable, its use does need to be carefully thought through, with emphasis on diversification of resistance mechanisms, a combination of both broad spectrum and narrow spectrum resistance QTL, and the implementation of certain cultural practices, such as cultivar mixtures for example.

### 4.4. Molecular Mechanisms Underlying Resistance QTL

Identifying the genes underlying the detected QTL would provide more insight into quantitative resistance and reveal the variety of mechanisms potentially involved in resistance to *Z. tritici*. Moreover, the use in breeding of resistance factors with distinct mechanisms could yet more complexify any pathogen adaptation [76,87].

For the three QTL detected in this study, the parent carrying the resistance allele is the cultivar Renan. Renan carries known resistance genes to a number of diseases, including rust resistance genes *Yr17*, *Lr37*, and *Sr38*, all originating from a chromosomal introgression from *Aegilops ventricosa* on wheat chromosome 2AS [88,89,90], eyespot resistance gene *Pch1* on chromosome 7D [91,92], and powdery mildew resistance gene *Pm4b* on chromosome 2AL [91,93]. None of these known resistances to other diseases colocalize with the three QTL detected in this study. *Qstb-renan-7B* was detected on the long arm of chromosome 7B. Two *Stb* genes are found on chromosome 7B [15], these are *Stb13* and *Stb8*. *Qstb-renan-7B* could colocalize with *Stb8* as it was detected towards the telomeric region of the long arm of chromosome 7B although this should be validated by using microsatellite markers Xgwm146 and Xgwm577 [94]. Chromosome 1D carries two known *Stb* genes, these are *Stb10* and *Stb19* [15,16]. *Stb10* is however not found close to the telomere as *Qstb-renan-1D* is, but rather near the centromere [95]. This QTL does however colocalize with *Stb19* [16]. The twenty-seven candidate genes in the *Stb19* region identified by Yang et al. (2018) with known *R* gene families are included in the *Qstb-renan-1D* confidence interval [16]. Nine of these are in our NB-LRR type gene list (Table 6) and four of these *BST_chr1D_nlr_13*, *BST_chr1D_nlr_113*, *BST_pseudo_chr1D_nlr_112*, and *BST_chr1D_nlr_14* had expression in the case of a *Z. tritici* infection (Figure 5). Therefore, *Qstb-renan-1D* could be an allele at *Stb19* although this remains to be validated through an allelism test between cultivars Renan and Lorikeet, the cultivar from which *Stb19* was identified [16]. Chromosome 5D does not carry any known *Stb* genes although STB resistance QTL have been previously identified on the short arm of the chromosome. Bearing in mind that the *Qstb-renan-5D* confidence interval overlaps both the short and long arm, we can therefore not exclude the possibility that it colocalizes with previously identified QTL [18]. *Qstb-renan-5D* explained up to 36% of phenotypic variation and is located in a region where no *Stb* gene has been mapped before. As previously discussed, the percentage of explained phenotypic variation cannot be considered as a criterion for the designation of a new *Stb* gene, because this percentage will strongly vary depending on the combination of effective resistances against a particular strain [24]. We therefore propose to designate this locus *Stb20q* in accordance with the *Stb* nomenclature adding a ‘q’ to indicate the quantitative nature of this locus despite its strong effect on resistance.

The three QTL detected here span overall quite a large interval on the genetic map and contain large numbers of genes, it is therefore impossible to consider studying all of these one by one. It is also possible that the gene or genes underlying the QTL are not present in Chinese Spring, it could therefore be interesting to look at other cultivar annotations. Renan in particular would be pertinent and the data will be available soon (F. Choulet, personal communication). The next best option would be to prune the list of genes to look into. Here, we chose to look into WAK, NB-LRR type and kinase domain carrying genes present in the QTL regions as these are seemingly the most likely gene families involved in quantitative disease resistance [6,7,10,14] in the context of a potential gene-for-gene interaction [23,24,96], a hypothesis we are encouraged to support by the strain specificities of the identified QTL. We are able to identify a short-list of genes in these gene families which could explain the detected QTL based on expression data, however, it would be interesting to test these further in the I05/Renan or I07/Renan interactions as the RNAseq data we have available was acquired from Riband inoculated with IPO323 [58]. So far, our results do not exclude that the genes underlying the detected resistance QTL could be based on similar mechanisms as known major *R* genes. However, other as yet unexplored possibilities cannot be ignored, such as those suggested by Poland et al. [87] who proposed a variety of possibilities, including host plant development or morphology regulating genes, mutations in basal defence genes, detoxification mechanisms, defence signal transduction or partially altered weaker forms of *R* genes. Their last suggestion was that the genes underlying resistance QTL could be a unique set of previously unidentified genes.

## 5. Conclusions

This study showed that the resistance in Renan to STB is quantitative and polygenic. In particular, it showed that Renan has a resistance QTL with a small effect, which colocalizes with *Stb19* and a QTL with a strong effect on chromosome 5D, which was designated as *Stb20q*. For breeding, it could be interesting to introgress these regions from Renan into wheat cultivars as they would, in theory, be able to hold up resistance after pyramided major resistance genes are overcome, or even limit the erosion of major resistance genes [76]. Nevertheless, the QTL were detected with data generated in controlled conditions and at seedling stage, they should be validated in field conditions before any possible implementation in breeding programs. It seems the next step would be to fine-map the QTL intervals, as this would give us a better idea as to which of the identified candidate genes, if any, are responsible for the resistance QTL. Furthermore, we are currently investigating pathogenicity QTL in our strains of interest I05 and I07, in order to evidence potential interactions between known resistance QTL in Renan and fungal QTL, with the aim of deciphering the mechanisms of minor gene-for-minor gene interactions.

## Figures and Tables

**Figure 1 genes-13-00100-f001:**
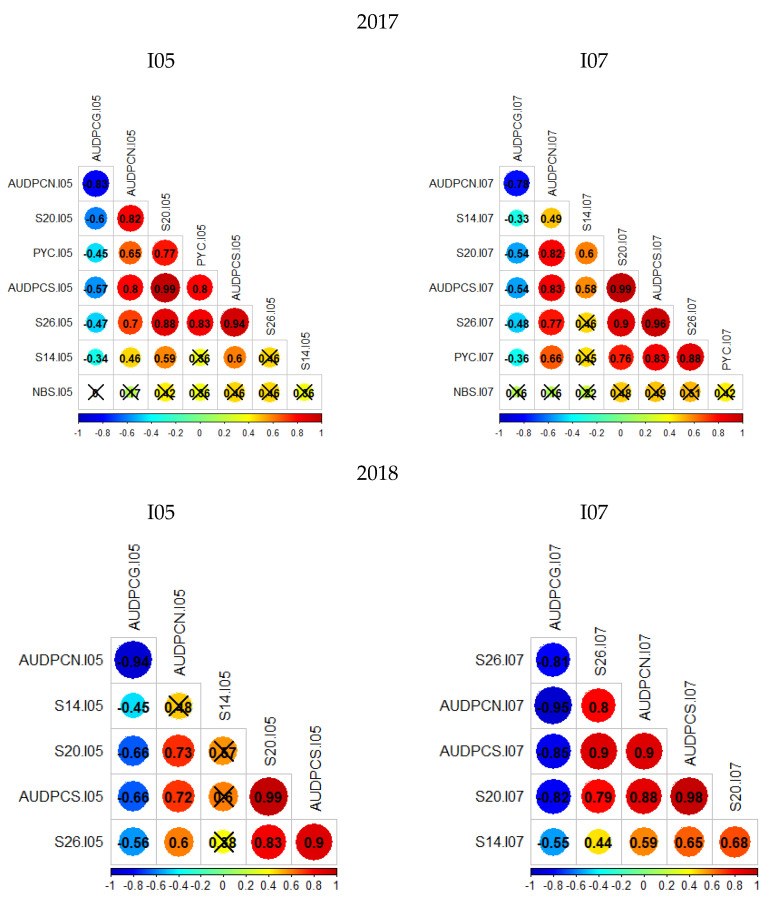
Bravais–Pearson correlograms for the four datasets; crossed out correlation values are not statistically significant (*p*-value < 0.05). The left column presents the phenotypic trait correlations for RxCS inoculated with I05, the right presents the phenotypic trait correlations for RxCS inoculated with I07. PYC is the pycnidia density. NBS is the number of spores per pycnidiospore. AUDPCG is the area under the disease progress curve for the green leaf area. AUDPCN is the area under the disease progress curve for the necrotic leaf area. AUDPCS is the area under the disease progress curve for the sporulating leaf area. S14, S20, and S26 are the sporulating leaf area at 14, 20, and 26 days post-inoculation respectively.

**Figure 2 genes-13-00100-f002:**
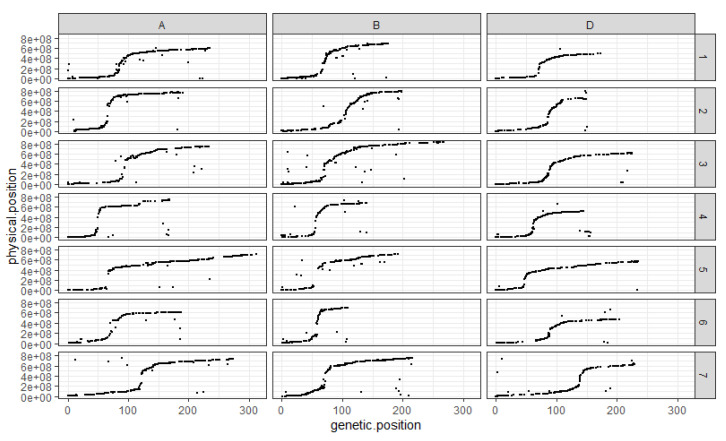
Comparison between markers’ genetic and assumed physical positions for each chromosome in each sub-genome. The *x*-axis corresponds to the genetic position of the mapped SNP on the RxCS genetic map in cM. The *y*-axis corresponds to the assumed physical position of the SNP on the chromosome they are on in bp. Each column corresponds to a wheat sub-genome, A, B and D respectively. Each line corresponds to a chromosome number; the chromosomes are numbered 1 through 7. Each black dot corresponds to a SNP.

**Figure 3 genes-13-00100-f003:**
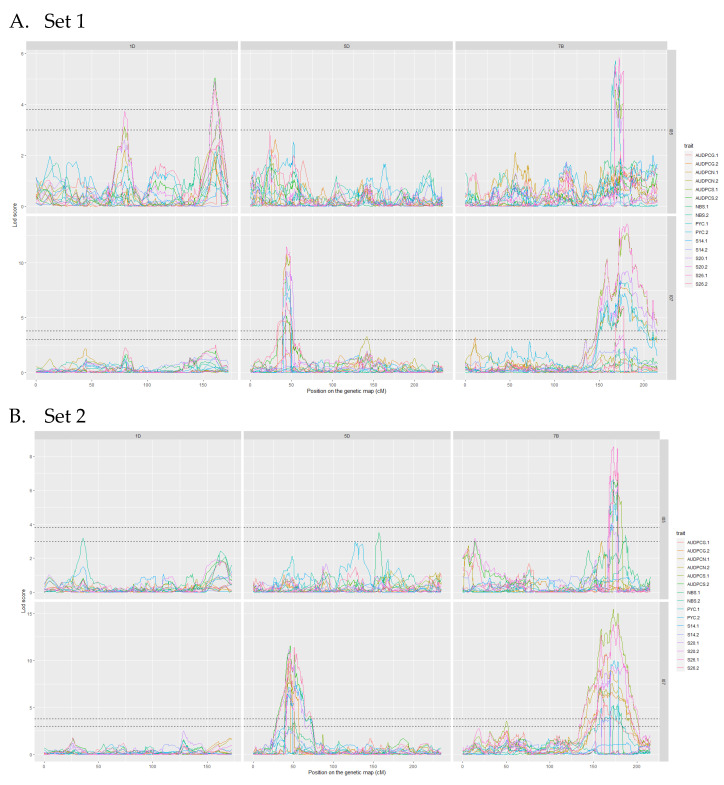
LOD score profiles for the linkage analyses of the 2017 datasets. (**A**) presents the results for Set 1, (**B**) presents the results for Set 2. In dotted lines are represented the minimal and maximal LOD threshold values obtained in the linkage analyses after 1000 permutations tests. The *x*-axis represents the position of the markers on the genetic map in cM. The *y*-axis represents the LOD score associated to the markers. Each column corresponds to a chromosome, chromosomes 1D, 5D and 7B respectively. Each line corresponds to a fungal strain, I05 and I07 respectively. The colours in the graphs correspond to the studied traits for each replication. PYC is the pycnidia density. NBS is the number of spores per pycnidiospore. AUDPCG is the area under the disease progress curve for the green leaf area. AUDPCN is the area under the disease progress curve for the necrotic leaf area. AUDPCS is area under the disease progress curve for the sporulating leaf area. S14, S20 and S26 are the sporulating area at 14, 20 and 26 days post-inoculation respectively. Each trait was studied over two replicates, 1 and 2.

**Figure 4 genes-13-00100-f004:**
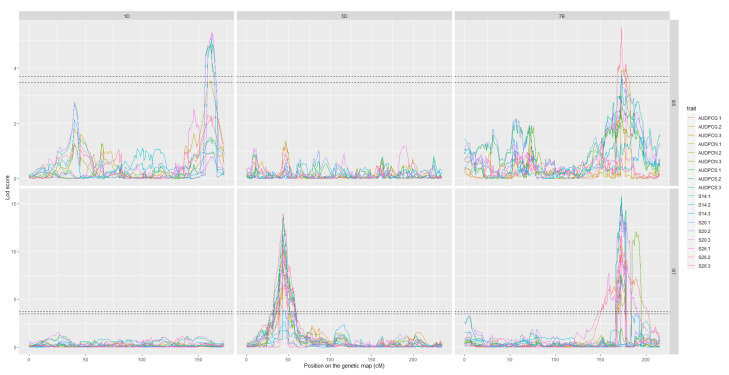
LOD score profiles for the linkage analyses of the 2018 dataset. In dotted lines are represented the minimal and maximal LOD threshold values obtained in the linkage analyses after 1000 permutations tests. The *x*-axis represents the position of the markers on the genetic map in cM. The *y*-axis represents the LOD score associated to the markers. Each column corresponds to a chromosome, chromosomes 1D, 5D, and 7B respectively. Each line corresponds to a fungal strain, I05 and I07 respectively. The colours in the graphs correspond to the studied traits for each replication. AUDPCG is the area under the disease progress curve for the green leaf area. AUDPCN is the area under the disease progress curve for the necrotic leaf area. AUDPCS is area under the disease progress curve for the sporulating leaf area. S14, S20, and S26 are the sporulating area at 14, 20, and 26 days post-inoculation respectively. Each trait was studied over three replicates, 1, 2, and 3.

**Figure 5 genes-13-00100-f005:**
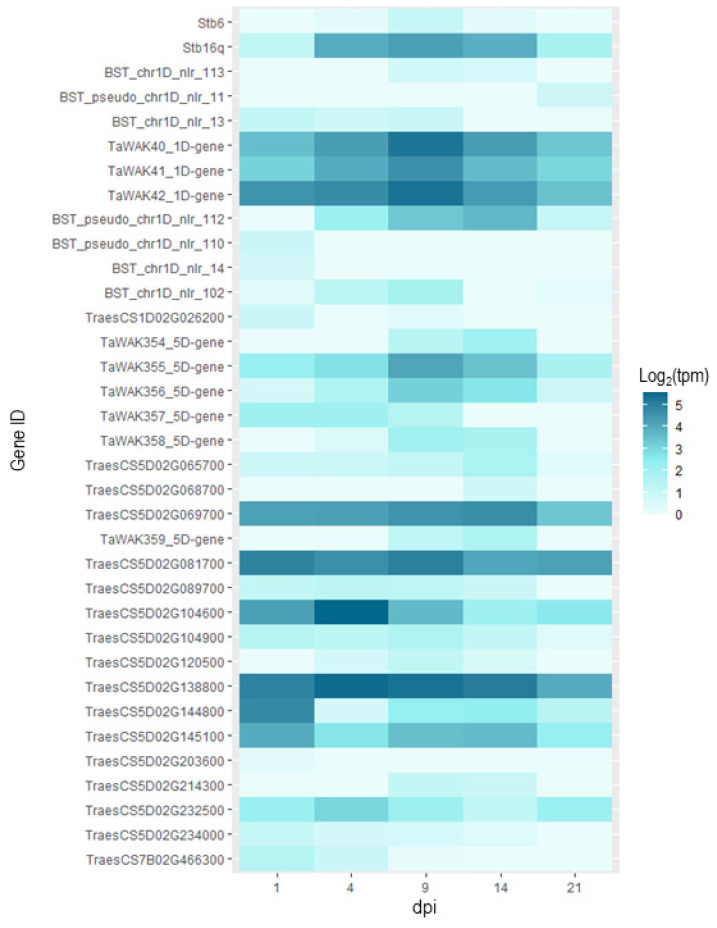
Heatmap representing the expression for each candidate gene where expression values were not equal to zero for the cultivar Riband inoculated with fungal strain IPO323. The *x*-axis represents post inoculation days, 1, 4, 9, 14, and 20 dpi respectively. The expression data is expressed in log_2_(tpm), with tpm transcripts per million. The first two lines present the data for *Stb6* and *Stb16q* for reference. The subsequent lines represent the candidate genes, which are ordered by physical position, top to bottom. The data represented here was extracted from http://www.wheat-expression.com/; accessed on 3 December 2021.

**Table 1 genes-13-00100-t001:** Settings for the Occhio Flowcell FC200S+HR.

Particle Counter	Diameter	0–8 µm
	Size	7–85 µm
	Circonference	0–0.70 µm
Grey value		195–205
Luminous intensity		7.5
Spacer thickness		150 µm
Resolution	Magnification	×4
	Calibration	0.47 µm.pixel−1 (1 pixel = 1.67 µm)

**Table 2 genes-13-00100-t002:** Statistical analysis of the phenotypic data for each dataset.

Isolate Set	Trait	Statistical Significance of the Genotype ^1^	Statistical Significance of the Replication ^1^	MSg ^2^	MSε ^3^	Broad-Sense Heritability	Shapiro–Wilk Normality Test on Residuals	Independence of Residuals	Homoscedasticity Bartlett Test of Homogeneity of Variances
I05_2017/1	S14		***	27	27	0.00	0.00	no	0.04
	S20	***	***	1094	444	0.42	0.11	yes	0.14
	S26	***	***	877	364	0.41	0.12	yes	0.03
	AUDPCG	*	*	51,787	35,921	0.18	0.92	yes	0.05
	AUDPCN	*	*	77,482	55,907	0.16	0.85	yes	0.31
	AUDPCS	***	***	83,562	32,115	0.44	0.11	yes	0.26
	PYC	***		21,769	9037	0.41	0.29	yes	0.29
	NBS		***	1,003,915	868,109	0.07	0.00	yes	0.05
I07_2017/1	S14		**	2	2	0.04	0.00	no	0.00
	S20	***	***	401	157	0.44	0.00	yes	0.85
	S26	***	***	765	233	0.53	0.40	yes	0.98
	AUDPCG	***		57,879	30,256	0.31	0.00	no	0.00
	AUDPCN	***	***	92,012	25,943	0.56	0.95	yes	0.77
	AUDPCS	***	***	40,045	12,909	0.51	0.00	yes	0.97
	PYC	***		29,153	8834	0.53	0.01	no	0.04
	NBS	***		1,865,055	630,731	0.49	0.00	no	0.00
I05_2017/2	S14	***	*	14	7	0.34	0.00	no	0.00
	S20	**	.	533	344	0.22	0.94	yes	0.54
	S26	**		665	378	0.28	0.40	yes	0.89
	AUDPCG	*	***	38,627	27,765	0.16	0.02	no	0.04
	AUDPCN	*	***	62,573	42,711	0.19	0.93	yes	0.35
	AUDPCS	**	.	44,817	26,387	0.26	0.74	yes	0.79
	PYC	***	***	22,651	12,737	0.28	0.97	yes	0.34
	NBS	**	***	816,528	495,995	0.24	0.00	yes	0.04
I07_2017/2	S14	***		31	13	0.42	0.00	no	0.00
	S20	***		1195	189	0.73	0.00	no	0.01
	S26	***		1411	222	0.73	0.60	yes	0.33
	AUDPCG	***		79,008	28,924	0.46	0.16	yes	0.22
	AUDPCN	***		115,168	23,787	0.66	0.24	yes	0.01
	AUDPCS	***		102,594	13,479	0.77	0.00	no	0.03
	PYC	***	***	59,415	18,267	0.53	0.82	yes	0.65
	NBS	*	*	1,341,693	906,931	0.19	0.00	yes	0.27
I05_2018 only replications 1 and 3	S14	**	***	140	86	0.24	0.00	no	0.00
	S20	***	***	1649	747	0.38	0.28	yes	0.94
	S26	*	***	877	661	0.14	0.00	no	0.82
	AUDPCG	***	***	32,787	17,602	0.30	0.00	no	0.00
	AUDPCN	***	***	58,506	28,592	0.34	0.86	yes	0.26
	AUDPCS	***	***	115,164	50,508	0.39	0.66	yes	0.65
I07_2018	S14	***		43	22	0.25	0.00	no	0.00
	S20	***	**	3061	368	0.71	0.00	no	0.00
	S26	***	***	3270	477	0.66	0.00	no	0.00
	AUDPCG	***		144,099	32,073	0.54	0.32	yes	0.02
	AUDPCN	***	**	170,267	29,795	0.61	0.00	yes	0.00
	AUDPCS	***	***	242,232	25,587	0.74	0.01	yes	0.00

^1^ Significance codes: 0 ‘***’, 0.001 ‘**’, 0.01 ‘*’, 0.05 ‘.’, 0.1 ‘ ’, 1; ^2^ MSg is the mean square value for the individual genotypes output by the ANOVA; ^3^ MSε is the mean square value for the residuals output by the ANOVA.

**Table 3 genes-13-00100-t003:** Characteristics of the RxCS genetic linkage map.

Chromosome		Number of SNP	Number of Genetic Bins	Map Length (cM)	Marker Density (Markers/cM)
A—genome	1	10,479	358	234.44	44.7
	2	11,716	288	189.72	61.8
	3	8413	302	233.28	36.1
	4	8465	203	170.24	49.7
	5	6723	411	312.17	21.5
	6	7816	226	189.03	41.3
	7	10,051	378	273.38	36.7
total A		63,663	2167	1602.26	39.7
B—genome	1	10,503	309	176.4	59.5
	2	9158	298	197.98	46.3
	3	12,016	374	268.03	44.8
	4	5888	180	140.45	41.9
	5	5019	234	192.98	26
	6	10,210	173	109.66	93.1
	7	7803	336	215.35	36.2
total B		60,597	1905	1300.85	46.6
D—genome	1	3609	157	172.81	20.9
	2	4601	134	150.27	30.6
	3	3512	225	225.25	15.6
	4	2131	174	156.99	13.6
	5	2581	203	234.75	11
	6	4180	183	204.6	20.4
	7	3946	211	228.97	17.2
total D		24,560	1287	1373.64	17.9
Total		148,820	5357	4276.75	34.8

**Table 4 genes-13-00100-t004:** QTL for resistance to STB detected with the phenotypic data generated in 2017 and the RxCS genetic map.

QTL.2017	Number of Detections	r^2^ Max (%)	Mean r^2^ (%)	Peak Marker Associated with r^2^ Max	Parent Carrying the Resistance Allele	Traits	Detected with
*Qstb.renan-1D*	3	7.5	6	cfn1317667_410K_1DS	Renan	S20, S26, AUDPCS	I05
*Qstb.renan-5D*	22	35.5	26	cfn2823104_410K_5DS	Renan	S20, S26, AUDPCG, AUDPCN, AUDPCS, PYC	I07
*Qstb.renan-7B*	37	32	20	cfn0449267_410K_7BL	Renan	S14, S20, S26, AUDPCG, AUDPCN, AUDPCS, PYC, NBS	I05 and I07

**Table 5 genes-13-00100-t005:** QTL for resistance to STB detected with the phenotypic data generated in 2018 and the RxCS genetic map.

QTL.2018	Number of Detections	r^2^ Max (%)	Mean r^2^ (%)	Peak Marker Associated with r^2^ Max	Parent Carrying the Resistance Allele	Traits	Detected with
*Qstb.renan-1D*	5	15	13.5	cfn1315024_410K_1DS	Renan	S20, AUDPCN, AUDPCS	I05
*Qstb.renan-5D*	18	21.5	15.5	cfn2827993_410K_5DS	Renan	S14, S20, S26, AUDPCG, AUDPCN, AUDPCS	I07
*Qstb.renan-7B*	22	38	21	cfn0916416_410K_7BL	Renan	S14, S20, S26, AUDPCG, AUDPCN, AUDPCS	I05 and I07

**Table 6 genes-13-00100-t006:** WAK ^1^, NB-LRR ^2^, and kinase domain carrying genes identified in the three QTL intervals.

QTL	Gene.ID	RefSeq v1.1 ID	Start (bp)	Stop (bp)	Annotation
*Qstb-renan-1D*	*BST_chr1D_nlr_115*	*TraesCS1D02G015500*	7277369	7280463	NB-LRR
*Qstb-renan-1D*	*BST_chr1D_nlr_114*	*TraesCS1D02G016026*	7381284	7384806	NB-LRR
*Qstb-renan-1D*	*BST_chr1D_nlr_113*	*TraesCS1D02G016100*	7419157	7422949	NB-LRR
*Qstb-renan-1D*	*BST_chr1D_nlr_9*	*TraesCS1D02G016900*	7592690	7609204	NB-LRR
*Qstb-renan-1D*	*BST_pseudo_chr1D_nlr_10*	*TraesCS1D02G016983*	7671168	7676063	NB-LRR
*Qstb-renan-1D*	*BST_pseudo_chr1D_nlr_11*	*TraesCS1D02G016991*	7678267	7680938	NB-LRR
*Qstb-renan-1D*	*BST_chr1D_nlr_12*	*TraesCS1D02G017400*	7820918	7823449	NB-LRR
*Qstb-renan-1D*	*BST_chr1D_nlr_13*	*TraesCS1D02G017600*	7868467	7872465	NB-LRR
*Qstb-renan-1D*	*BST_pseudo_chr1D_nlr_112*	*TraesCS1D02G018700*	8182627	8186066	NB-LRR
*Qstb-renan-1D*	*BST_expressed_pseudo_chr1D_nlr_111*	*TraesCS1D02G018800*	8226547	8230158	NB-LRR
*Qstb-renan-1D*	*BST_pseudo_chr1D_nlr_110*	*TraesCS1D02G019600*	8605145	8610344	NB-LRR
*Qstb-renan-1D*	*BST_chr1D_nlr_14*	*TraesCS1D02G019700*	8610887	8616204	NB-LRR
*Qstb-renan-1D*	*BST_pseudo_chr1D_nlr_109*	*TraesCS1D02G020619*	8838803	8842102	NB-LRR
*Qstb-renan-1D*	*BST_chr1D_nlr_108*	*TraesCS1D02G021000*	9028322	9037195	NB-LRR
*Qstb-renan-1D*	*BST_pseudo_chr1D_nlr_106*	*TraesCS1D02G021200*	9086119	9091764	NB-LRR
*Qstb-renan-1D*	*BST_pseudo_chr1D_nlr_104*	*TraesCS1D02G021751*	9309301	9328352	NB-LRR
*Qstb-renan-1D*	*BST_chr1D_nlr_16*	*TraesCS1D02G022500*	9575753	9593333	NB-LRR
*Qstb-renan-1D*	*BST_chr1D_nlr_102*	*TraesCS1D02G026000*	10661025	10664946	NB-LRR
*Qstb-renan-1D*	*BST_chr1D_nlr_17*	*TraesCS1D02G028200*	11175841	11182532	NB-LRR
*Qstb-renan-1D*	*BST_chr1D_nlr_18*	*TraesCS1D02G028600*	11272449	11278362	NB-LRR
*Qstb-renan-1D*	*BST_expressed_pseudo_chr1D_nlr_19*	*TraesCS1D02G028700*	11287245	11292876	NB-LRR
*Qstb-renan-1D*	*BST_pseudo_chr1D_nlr_20*	*TraesCS1D02G028736*	11319796	11321348	NB-LRR
*Qstb-renan-1D*	*BST_chr1D_nlr_21*	*TraesCS1D02G029000*	11408761	11415088	NB-LRR
*Qstb-renan-1D*	*BST_chr1D_nlr_22*	*TraesCS1D02G029100*	11451423	11459353	NB-LRR
*Qstb-renan-1D*	*BST_chr1D_nlr_23*	*TraesCS1D02G029200*	11493627	11499140	NB-LRR
*Qstb-renan-1D*	*TaWAK38_1D-gene*	*TraesCS1D02G016200*	7429822	7445013	WAK
*Qstb-renan-1D*	*TaWAK39_1D-gene*	*TraesCS1D02G016800*	7583590	7587977	WAK
*Qstb-renan-1D*	*TaWAK40_1D-gene*	*TraesCS1D02G017700*	7874518	7876881	WAK
*Qstb-renan-1D*	*TaWAK41_1D-gene*	*TraesCS1D02G017800*	7877418	7880329	WAK
*Qstb-renan-1D*	*TaWAK42_1D-gene*	*TraesCS1D02G017900*	7896854	7899155	WAK
*Qstb-renan-5D*	*TaWAK349_5D-gene*	*TraesCS5D02G043400*	42925913	42928461	WAK
*Qstb-renan-5D*	*TaWAK350_5D-gene*	*TraesCS5D02G043500*	42930408	42932902	WAK
*Qstb-renan-5D*	*TaWAK351_5D-gene*	*TraesCS5D02G043532*	42944893	42947212	WAK
*Qstb-renan-5D*	*TaWAK352_5D-gene*	*TraesCS5D02G052500*	50569632	50576495	WAK
*Qstb-renan-5D*	*TaWAK353_5D-gene*	*TraesCS5D02G052800*	50635242	50646756	WAK
*Qstb-renan-5D*	*TaWAK354_5D-gene*	*TraesCS5D02G061800*	58138943	58142318	WAK
*Qstb-renan-5D*	*TaWAK355_5D-gene*	*TraesCS5D02G061900*	58143379	58149665	WAK
*Qstb-renan-5D*	*TaWAK356_5D-gene*	*TraesCS5D02G062100*	58151124	58155524	WAK
*Qstb-renan-5D*	*TaWAK357_5D-gene*	*TraesCS5D02G062200*	58226914	58230221	WAK
*Qstb-renan-5D*	*TaWAK358_5D-gene*	*TraesCS5D02G062600*	58419864	58422609	WAK
*Qstb-renan-5D*	*TaWAK359_5D-gene*	*TraesCS5D02G073900*	72901902	72907097	WAK
*Qstb-renan-5D*	*TaWAK360_5D-gene*	*TraesCS5D02G096200*	106519841	106525422	WAK
*Qstb-renan-7B*	*TaWAK556_7B-gene*	*TraesCS7B02G463200*	720131495	720134235	WAK
*Qstb-renan-1D*	*TraesCS1D02G026200*	*TraesCS1D02G026200*	10715309	10722269	Probable serine/threonine-protein kinase WNK3
*Qstb-renan-5D*	*TraesCS5D02G060900*	*TraesCS5D02G060900*	57843934	57851315	Non-specific serine/threonine protein kinase
*Qstb-renan-5D*	*TraesCS5D02G065700*	*TraesCS5D02G065700*	61052683	61060726	Phosphatidylinositol 3-kinase VPS34
*Qstb-renan-5D*	*TraesCS5D02G068700*	*TraesCS5D02G068700*	65753632	65755248	Non-specific serine/threonine protein kinase
*Qstb-renan-5D*	*TraesCS5D02G069700*	*TraesCS5D02G069700*	67578001	67588803	pfkB-like carbohydrate kinase family protein
*Qstb-renan-5D*	*TraesCS5D02G081700*	*TraesCS5D02G081700*	82186877	82189457	Serine/threonine protein kinase%2C Abscisic acid (ABA)-activated protein kinase%2C Hyperosmotic stress response%2C ABA signal transduction
*Qstb-renan-5D*	*TraesCS5D02G089700*	*TraesCS5D02G089700*	97036711	97041067	Diacylglycerol kinase
*Qstb-renan-5D*	*TraesCS5D02G091000*	*TraesCS5D02G091000*	98227410	98230028	L-type lectin-domain containing receptor kinase S.4
*Qstb-renan-5D*	*TraesCS5D02G104600*	*TraesCS5D02G104600*	118455172	118460088	Nucleoside diphosphate kinase
*Qstb-renan-5D*	*TraesCS5D02G104900*	*TraesCS5D02G104900*	118834967	118838504	ATP-dependent 6-phosphofructokinase
*Qstb-renan-5D*	*TraesCS5D02G120500*	*TraesCS5D02G120500*	170376901	170381844	Diacylglycerol kinase
*Qstb-renan-5D*	*TraesCS5D02G138800*	*TraesCS5D02G138800*	221007037	221012985	Pyruvate kinase
*Qstb-renan-5D*	*TraesCS5D02G140700*	*TraesCS5D02G140700*	224325320	224328774	Phosphatidylinositol 4-phosphate 5-kinase
*Qstb-renan-5D*	*TraesCS5D02G144800*	*TraesCS5D02G144800*	231350992	231353762	Non-specific serine/threonine protein kinase
*Qstb-renan-5D*	*TraesCS5D02G145100*	*TraesCS5D02G145100*	231743581	231750603	Mitogen-activated protein kinase
*Qstb-renan-5D*	*TraesCS5D02G166400*	*TraesCS5D02G166400*	259233864	259236422	Receptor like protein kinase S.2
*Qstb-renan-5D*	*TraesCS5D02G181500*	*TraesCS5D02G181500*	282151742	282156543	BR receptor kinase%2C Brassinosteroid (BR) perception in the roo
*Qstb-renan-5D*	*TraesCS5D02G191900*	*TraesCS5D02G191900*	294637785	294639948	NAD(H) kinase 3
*Qstb-renan-5D*	*TraesCS5D02G203600*	*TraesCS5D02G203600*	308863403	308865114	Serine/threonine-protein kinase BLUS1
*Qstb-renan-5D*	*TraesCS5D02G214300*	*TraesCS5D02G214300*	323911872	323914256	Serine/threonine-protein kinase
*Qstb-renan-5D*	*TraesCS5D02G232500*	*TraesCS5D02G232500*	339652596	339654075	Non-specific serine/threonine protein kinase
*Qstb-renan-5D*	*TraesCS5D02G232600*	*TraesCS5D02G232600*	339712134	339713477	Non-specific serine/threonine protein kinase
*Qstb-renan-5D*	*TraesCS5D02G234000*	*TraesCS5D02G234000*	341192646	341202493	ATP-dependent 6-phosphofructokinase
*Qstb-renan-7B*	*TraesCS7B02G466300*	*TraesCS7B02G466300*	723900282	723903056	Serine/threonine-protein kinase

^1^ Wall-associated kinases (WAK); ^2^ Nucleotide-binding and leucine-rich repeats (NB-LRR); Genes highlighted are the candidate genes found in common with Yang et al. (2018) [16].

## Data Availability

The ultra-dense genetic map built from the population Renan x Chinese Spring and used to map resistance QTL is provided as a Appendix A file.

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
