# Peer review of "Resistance of the Wheat Cultivar ‘Renan’ to Septoria Leaf Blotch Explained by a Combination of Strain Specific and Strain Non-Specific QTL Mapped on an Ultra-Dense Genetic Map"

_genes, 2021, doi:10.3390/genes13010100_

Round 1

Reviewer 1 Report

The manuscript by Langlands-Perry et al. reported the ultra high density SNP genetic linkage map construction and QTL mapping of Septoria leaf blotch resistance using a RIL population derived from Renan / Chinese Spring. The authors developed an linkage map containing 148820 SNP markers composing 5357 genetic bins using two wheat SNP arrays (TaBW410K and iSelect90K). Phenotypic data collected in two cropping seacons for reaction to two Zymoseptoria tritici isolates were used for mapping QTL responsible for STB resistance. Major effects QTL loci were identified on chromosomes 7B, 1D and 5D. The 5D QTL was novel and designated as Stb20q. These results are informative useful for wheat researchers and breeders for identifying other interested traits using the Renan / CS linkage map and breeding STB resistance cultivars using the QTL in marker assisted selection.

However, I have a major comment regarding to the claims by the authors for the durability of quantitative resistance, not qualitative resistance in this manuscript. For the QTL identified by the authors, the 7B and 1Dloci are indeed STb8 and Stb19, which are major genes or qualitative resistance. The novel one on 5D may be aslo treated as a major locus. Meantime, the authors looked through the 5D locus mapping interval to find WAK, NBS-LRR, and Kinase proteins which are mostly fit the (reverse) gene-for-gene interaction relationship. I would suggest the authors not emphasis too much about the difference of the two kinds of resisatance, but focused on the ideitification, mapping and potential value of the QTL loci they found. As we know, many of the QTL are indeed single genes when the genetic background is simplified enough.    

Author Response

Dear Madam or Sir,

Thank you for reviewing our work.​

We understand what you mean with regards to the nature of the resistance studied here and do agree that a single gene can underlie a QTL. However, we stand by the denomination quantitative resistance, and this for several reasons. Firstly, the most basic, looking at the distribution of phenotypes, we observe continuous distribution, not discrete as would be expected in a qualitative context. Secondly, as you rightly say, the 1D locus does indeed seem to be Stb19, however the QTL only explains up to 15% of phenotypic variation, which can hardly be considered major. Moreover, though the 7B and 1D QTL colocalize with major resistance genes, we cannot say with certainty that they are Stb8 and Stb19. Our results show that it is the contribution of each of the loci that leads to an overall resistant phenotype, meaning that it is several genes with combined effects which contribute, and therefore the system is polygenic. All this points towards quantitative resistance.

We do agree with you on the fact that WAK, NB-LRR and kinase proteins fit the gene-for-gene interaction relationship, however, these types of genes have also been identified in a quantitative context in Fusarium, rice blast and a number of other organisms which are referred to in the introduction. What we wish to show with later studies, as hinted in the conclusion, is that quantitative and qualitative interactions are not so different, and that the genes involved in quantitative interactions are similar to those involved in qualitative interactions and that "quantitative" and "gene-for-gene interaction" are not mutually exclusive. For these reasons, we feel justified in keeping the emphasis on quantitative.

To justify the choice of genes looked into despite a quantitative context, we propose to modify the text as such:

Line 700: “Here, we chose to look into WAK, NB-LRR type and kinase domain carrying genes present in the QTL regions as these are seemingly the most likely gene families involved in quantitative disease resistance [6,7,10,14] and in the context of a potential gene-for-gene interaction [96,23,24], a hypothesis we are encouraged to support by the strain specificities of the identified QTL.”

Thank you very much for your review,

Camilla Langlands-Perry

Reviewer 2 Report

Comment: Minor revision

This research is very interesting and highly impactful in the area of plant protection especially the management of Septoria leaf blotch disease in wheat. The author’ has done a great job but still, some issues need to be addressed. The comments for the authors are provided below:

The scientific name should be in italic. Correct it in the whole manuscript.

Line20: please, make clear ‘three resistance QTL’ or ‘three QTLs related to resistance’.

Line: 105: place a comma (,) after the word ‘Hereafter’.

Line: 105: place a comma (,) after ‘I07’.

  1. Materials and methods: use appropriate references for each of the methods you followed.

Section 2.3.1: Provide appropriate references for the techniques used for the visual evaluation of symptoms.

Line 308-312: What is the cause of infection failure? Include in the text. Do you think that the results of the two replications reflect the actual results?

Author Response

Dear Madam or Sir,

Thank you for reviewing our work.

All of your comments have been taken into account but the one concerning lines 308-312. The reason for this exclusion is that we are not able to say exactly what went wrong with this experiment. The issue could be from the climate chamber or from the inoculum, however no issues with the climate chamber were reported for the experiment period. We therefore can suppose that the issue came from the inoculum, possibly from an issue with spore viability, inoculum conservation, etc. However, we cannot pinpoint one particular issue, this type of problem is rare but does happen, it is generally not explainable.

Thank you very much for your review,

Camilla Langlands-Perry